# Hibernation Patterns of the European Hedgehog, *Erinaceus europaeus*, at a Cornish Rescue Centre

**DOI:** 10.3390/ani10081418

**Published:** 2020-08-14

**Authors:** Kathryn E. South, Kelly Haynes, Angus C. Jackson

**Affiliations:** 1Prickles and Paws Hedgehog Rescue, Cubert, Newquay, Cornwall TR8 5HD, UK; 2Centre for Applied Zoology, Cornwall College Newquay, Newquay, Cornwall TR7 2LZ, UK; kelly.haynes@cornwall.ac.uk (K.H.); angus.jackson@cornwall.ac.uk (A.C.J.)

**Keywords:** hedgehog, hibernation, spontaneous arousal, metabolism, wildlife rehabilitation, rehabilitation protocols, wildlife rescue

## Abstract

**Simple Summary:**

Populations of the European hedgehog, *Erinaceus europaeus*, are declining in the UK. This small mammal is frequently admitted to rescue centres in the UK to be treated for a variety of illnesses or injuries. With many spending the winter in captivity, clear guidelines about how to look after hedgehogs during their hibernation would be very useful. We studied 35 hedgehogs over two winters to learn about their sleeping behaviour and how they change weight. We measured the total length of hibernation and the periods during hibernation when hedgehogs are more active (called spontaneous arousals). There were three main results. (1) The longer the hibernation, the more weight was lost. (2) Previous studies show that arousal is energetically expensive. Despite this, weight-loss was more related to the amount of time spent sleeping than to the number of times the hedgehog woke up, perhaps because they could easily feed each time they woke up. (3) Larger hedgehogs lost proportionally less weight per day, perhaps because they woke up and fed more often than did smaller hedgehogs. Behaviour by hibernating hedgehogs in captivity differs from that in the wild. Patterns revealed in this study are used to make some recommendations for guidelines that can be adapted for individual hedgehogs according to their size and behaviour during hibernation.

**Abstract:**

The European hedgehog, *Erinaceus europaeus*, is frequently admitted to rescue centres in the UK. With many overwintering in captivity, there is cause to investigate hibernation patterns in order to inform and improve husbandry and monitoring protocols. Thirty-five hedgehogs were studied over two winters. Weight change during hibernation for the first winter was used to test for effects of disturbance on different aspects of hibernation, including total duration, frequency and duration of spontaneous arousals. There was no significant difference between the two winters for any of the four aspects studied. Significant positive correlations demonstrated that weight-loss increased with the duration of the hibernation period and with percent of nights spent asleep, but not with the number of arousal events. Thus, weight-loss appears more strongly associated with the proportion of time spent asleep than with the number of arousal events. This was surprising given the assumed energetic expense of repeated arousal and was potentially due to availability of food during arousals. In contrast with previous studies, larger hedgehogs lost less weight per day than did smaller hedgehogs. They also woke up more often (i.e., had more opportunities to feed), which may explain the unexpected pattern of weight-loss. Hibernatory behaviour in captivity differs from that in the wild, likely because of non-natural conditions in hutches and the immediate availability of food. This study provides a basis for further research into the monitoring and husbandry of hedgehogs such that it can be adapted for each individual according to pre-hibernation weight and behaviour during hibernation.

## 1. Introduction

The Western European hedgehog (*Erinaceus europaeus*) is the most common mammal species admitted to wildlife rescue centres across the UK [1]. Their rehabilitation and release have the potential to contribute to conservation of this declining species. Whilst there are challenges with assessment of population sizes [2,3], the most recent estimate is 522,000, a 66% reduction from the 1995 estimate [3]. Reasons for rescue and rehabilitation are numerous, including: injuries from pets or gardening activities, entrapment, car-strikes, poisoning from pesticides and parasitic burdening [4,5]. Many hedgehogs undergoing or following rehabilitation are overwintered or hibernate at rescue centres [4,5]. 

### 1.1. Hibernation

In the UK, hibernation (a period of greatly reduced activity, temperature, respiration and metabolism) [6] by wild hedgehogs, commonly occurs in outdoor nests (or hibernacula) between November and March. It is triggered primarily by consistently low temperatures (<8 °C) [7], although photoperiod, body-condition and food availability may also be involved [8]. Date and duration of hibernation periods can be influenced by climate, individual condition and sex [7,8,9]. 

For wild hedgehogs, mortality during hibernation can be a major component of overall mortality [7,9] with as few as 30% of young surviving their first winter. Other estimates of survival probabilities during hibernation are more favourable, ranging from a mean of 0.66 in southern Sweden [10] to 0.89 (*n* = 18) in Denmark [9] and even 1.0 in Denmark (*n* = 6) [11]. Survival of winter may not occur if an individual is in poor condition and/or has small body weight [5,9,12]. Estimates of percent loss of weight during hibernation vary even within a country. For instance, in Britain, mean (±s.e.) weight-loss has been estimated as 14.11 (±3.08)% [13] and as 0.2% of original body-weight (or 0.8–2.0 g day^−1^) being lost for each day spent in hibernation [14]. In southern Ireland, loss of body-weight was 17 (±0.53)% over hibernations lasting 148.9 (±0.5) days [15]. In Denmark, Jensen [11] recorded weight-loss during hibernation of 22.1 (±10.1)% for juveniles (during a mean of 79 days hibernation, *n* = 6) and 30.2 (±10.1)% for adult females (during a mean of 198 days, *n* = 3). Furthermore, also in Denmark, Rasmusson [9] measured juvenile weight-loss as 16 (±2.9)%. Weight-loss will depend on pre-hibernation weight and lighter individuals typically lose smaller percentages of their pre-hibernation mass than do larger individuals [11,16]. Where durations of hibernation vary among individuals, the % change in weight will also vary and so to make fair comparisons among individuals or studies, it may be more informative to consider change in weight day^−1^.

### 1.2. Spontaneous Arousals

All hibernating mammals exhibit brief awakenings called spontaneous or periodic arousals during hibernation, where normal temperatures are regained and activity-levels increase [17]. Such spontaneous arousal is energetically expensive, with up to 75% of total energy requirement during hibernation being associated with arousals [18]. Considering how energetically expensive arousal during hibernation appears to be, the exact function and effects remain poorly understood. Possible functions are speculated to involve recovery from physiological costs accrued during metabolic depression, which may include oxidative stress, reduced immunocompetence and neuronal damage [17]. In the wild, hedgehog activity during these periods of arousal varies considerably. Walhovd [19] observed multiple arousals during hibernation (indicated by spontaneous increases in nest temperature), but no departure from the nest. Other studies report that arousals involve changing nest, sometimes several times [8,9,11]. Yarnell et al. [13] observed hibernation of 57 British hedgehogs in the wild, recording between one and seven nest changes per hedgehog. The types of nest and nesting material used (e.g., compost heap, twigs, under a bush, under a building) also vary among individuals and within a hibernation [9,20,21]. Arousal duration in outdoor hibernacula can last between 34 and 44 h [19] and frequency of spontaneous arousal has been recorded at 2.9 events per month (*n* = 29 individuals) [22]. Other similar figures are reported, yet demonstrate a large range, with 3–15 days between arousals and a total arousal duration during hibernation of 12–18 days [19]. Similarly, Buckle [23] reported a range of between 5 and 15 days between arousals whilst Kristoffersson and Soivio [24] reported that the longest periods between arousals lasted from 10–13 days. 

Not all hedgehogs lose weight during hibernation. Of 10 radio-tracked hedgehogs in Copenhagen, five increased in weight during hibernation [9], almost certainly because of opportunities to feed during periods of arousal. If arousals are required regularly (for physiological reasons) then the number of arousals should increase with the duration of the total hibernation period. Different frequencies and durations of arousal may influence weight-loss, because feeding is possible during arousal and/or because arousal is energetically expensive. Evidence is contradictory for whether the number of arousals is positively or negatively correlated with weight-loss, as summarised nicely by Haigh et al. [15]. 

### 1.3. Threshold Mass to Survive Hibernation

Given all this variation, there is much discussion about the minimal body-weight that would allow a hedgehog to survive hibernation. From post-hibernation weights of 105 hedgehogs, Morris [25] estimated each individual’s pre-hibernation weight, by assuming a likely loss of 25% of their body-weight during hibernation. With a buffer of 50 g to ensure sufficient fat reserves to survive the unpredictable British winters, he suggested a minimal weight for otherwise healthy individuals would be 450 g. In rural Denmark, a greater body-weight of 513 g was needed to survive hibernation, likely due to the harsher winter conditions [11]. In Ireland, late juveniles could survive the winter with a pre-hibernation weight of 475 g or more [15]. The minimal body-weight needed to survive hibernation at present may differ from these estimates, due to regional variation in climate and the passage of time since publication with the effects of climatic change on winter temperatures [9,26]. 

The health of a hedgehog and its probability of surviving winter in the wild is almost certainly down to more than just its absolute weight. A large, skinny hedgehog may weigh the same as a small fat hedgehog, but the former is in poor condition and may be more prone to mortality over winter than the latter. For those concerned with over-winter rehabilitation of hedgehogs in captivity, knowledge of hedgehog condition and likely patterns of behaviour will influence decisions about treatment. For example, body-weight prior to hibernation and anticipated change in weight may determine whether individuals are released to hibernate in the wild, or whether they are to hibernate in captivity or if they should be kept warm, awake, feeding and undergoing any necessary veterinary treatment. At present, in the UK, a minimal body-weight of 550 [7] or 600 g [5,12] is used as the threshold below which individuals are not released from rescue centres before winter and instead allowed to gain weight and hibernate in the centre. 

### 1.4. Hibernation in Captivity

Conditions for hibernation in captivity are, however, clearly different to those in the wild. For instance, captive hedgehogs cannot move far during arousal, cannot change nest, have only one type of nest material and food is readily available in the enclosures. Handling during hibernation for weighing or health checks may also artificially induce arousal [8,27]. Whether or how captive conditions will affect hibernatory behaviours when compared to those in the wild is not certain. 

Given the large numbers of hedgehogs hibernating in rescue centres over winter, knowledge of how weight will be lost and variation in behaviours among individuals is required because they have implications for husbandry (specifically mid-hibernation health-checks, weighing and artificial arousal). Thus, relationships between pre-hibernation mass, spontaneous arousal frequency and mass-loss during hibernation at rescue centres need further exploration. This will help ensure that appropriate intervals for weighing (and associated disturbance) are used, minimising the risk of fatalities during hibernation and improving the success of rehabilitation and release.

We used data on weight and activity of healthy juvenile hedgehogs available from records of existing husbandry protocols kept by a specialist hedgehog rescue centre, to test the following sets of hypotheses. 

Hibernation with or without weighing: To test whether disturbance by weighing in mid-hibernation affected overall hibernation, we predicted that there would be a difference in mean total hibernation periods (nights), mean number of arousal events, mean duration of arousal events (nights) and mean number of nights sleep between arousal events between the two winters. 

Associations with total hibernation period: We predicted that there would be correlations between the total hibernation period and the number of arousal events; or with percentage of time spent in arousal. More specifically, we expected that the longer the hibernation, the more arousal events there would be (a positive correlation) and that there could be more or less time spent asleep (a two-tailed correlation).

Associations with percent weight-loss: We tested hypotheses that weight-loss would be associated with: total hibernation period, number of arousal events, and percentage of nights asleep. Our expectation was that weight-loss would increase with the length of hibernation (a positive correlation), but we had no particular prediction for how it would change with number of arousal events or the proportion of time spent asleep (two-tailed correlations) 

Associations with pre-hibernation weight: Finally, we tested whether pre-hibernation weight would be associated with: daily % weight-loss, actual weight-loss, total hibernation period, number of arousal events, mean number of nights awake per arousal event, and mean number of nights asleep between arousal events. In each case, we were uncertain about how the variables might change with pre-hibernation weight (i.e., all correlations were 2-tailed). 

## 2. Materials and Methods

The study was conducted on 35 healthy, juvenile hedgehogs in good condition (17 males, 18 females) undergoing rehabilitation at Prickles and Paws Hedgehog Rescue Centre, Cornwall, UK, over the 2015/2016 and 2016/2017 winters. The centre, which works closely with a local veterinary practice, is a registered charity which operates to the standards of, and admits hedgehogs from, the Royal Society for the Prevention of Cruelty to Animals (RSPCA). The centre does not hold a UK Home Office licence under the Animal (Scientific Procedures) Act 1986, and data were collected only through the normal, approved operating protocols of the centre. 

Individuals used in the study were admitted to the centre for a range of reasons and, if necessary, appropriate medical treatments, as prescribed by the centre’s veterinarian, were administered (Appendix A). Historically, many rescue centres have opted to keep rehabilitated hedgehogs over winter, rather than releasing them before spring, even if in full health [7], and this has been the approach adopted by Prickles and Paws. No individual was considered for use in this study and allowed to hibernate unless it fulfilled the conditions of the centre’s Hibernation Protocol (Appendix A) including having all medications completed, being >600 g in weight and in good body condition. To account for mass relative to size, a simple body-condition index specifically for hedgehogs was developed [28]. Testing the application of this index, Rasmussen et al. [9] recently found it to be of uncertain reliability due to the need for precise measurements, difficult to achieve on hedgehogs. In the present study, body condition was determined by visual assessment of body-shape; near-spherical shape when curled-up indicates good body-condition, a triangular rear-end indicates underweight condition [28]. Hedgehogs in the study were housed individually in outdoor hutches (floor area 0.31–0.49 m^2^), where illumination, food and nesting material (shredded newspaper), were standardised as much as possible. Paper, whilst different to natural nesting materials, is readily available, dust-free, widely used in British rescue centres and recommended by other authors [7]. 

Between November 2015 and March 2016, weight-change and hibernatory behaviour were monitored for 21 healthy individuals. Prior to hibernation, individuals were weighed at least weekly to the nearest g using digital scales. When individuals displayed signs of entering hibernation (reduced food-intake and dark green faeces) all handling was ceased. There are several clinical reasons for green faeces, but in otherwise healthy hedgehogs, they may be an indicator of incipient hibernation and, therefore, a fair cue for changes in husbandry. If they had not entered hibernation within the next three days, they would then be weighed again. Thus, the time between last recorded weight and the start of hibernation varied from 1 to 6 days. When in hibernation, hedgehogs were checked daily by viewing through a mesh door and spontaneous arousals were noted when there was evidence of recent disturbance in the hutch (e.g., spread of nesting material, presence of faeces, food disturbed), which indicated the individual had left the nest area. After 6 weeks in hibernation, individuals were weighed; if their mass had fallen below 400 g, then arousal was induced artificially. This mass was considered a minimal threshold to ensure survival of the remainder of hibernation if there were no further spontaneous arousals. Dried food for small carnivores (e.g., cat biscuits) was available throughout, but food consumption during arousal was not recorded.

To assess potential effects of mid-hibernation weighing, frequencies and durations of spontaneous arousals, but not mid- or end-hibernation weights, were recorded for 14 individuals during a second winter (December 2016 to March 2017). The sample-size for 2016/17 was smaller, due to fewer healthy individuals of adequate mass being at the centre. 

Entry into hibernation was defined as the first night that the individual did not leave the nest area (as determined by no disturbance in the hutch). The date of final arousal from hibernation was the first of five consecutive nights of activity outside the nest area. Total hibernation period was the number of nights between first entry into hibernation and final arousal (i.e., including nights in spontaneous arousal). A spontaneous arousal event constituted a series of consecutive nights spent active during the total hibernation period. Periods of spontaneous arousal began when an individual left the nest area and ended when the individual returned to the nest and no further disturbance was observed. The percentage time in spontaneous arousal was the total number of nights in spontaneous arousal as a percentage of the total hibernation period. The duration of an interval between arousals was the number of nights between the end of one arousal and the start of the next. Behaviours tending to indicate spontaneous rather than final arousal (e.g., minimal food intake and activity) meant that sometimes, several days elapsed before final arousal was certain. Thus, post-hibernation weight was recorded as soon as final arousal was confirmed (typically within 3 days and always within 6 days). 

Statistical analysis: Weight change was measured as percentage or actual weight-loss relative to pre-hibernation weight, with positive or negative values representing decreases or increases in weight, respectively. Spontaneous arousal events were counted, measured for duration (number of nights per event) or total duration (sum of all events per hedgehog converted to a percentage of the total hibernation period). For correlations, Shapiro-Wilk normality tests were used to assess frequency distributions of data. Where distributions of data were non-normal or where at least one variable was a percentage, Spearman’s rank correlation coefficients were used to establish significance of associations. When comparing variables between years, Fisher’s F-tests were used to test for heteroscedasticity. Where variances were homogeneous, Student’s *t*-test was used. All analyses were done in R version 4.0.2 [29]. 

## 3. Results 

### 3.1. Hibernation with or without Weighing

Mean total hibernation periods were of similar duration in 2015/16 and in 2016/17 (Figure 1, Table 1). Maximal total hibernation periods for the two winters were 111 nights (106:5 nights asleep:awake) and 70 nights (68:2), respectively. Minimal durations were 13 nights (9:4) and 5 nights (5:0). Spontaneous arousals were observed in all but one individual (which hibernated for only 5 nights). The most spontaneous arousal events for a single individual in 2015/16 was 15 and in 2016/17, three individuals displayed six arousals. The mean number of arousals did not differ between the two winters (Figure 1, Table 1). Some spontaneous arousals lasted multiple nights, the longest being 9 nights, but neither the mean number of nights per event nor the number of nights between events differed between the two winters (Figure 1, Table 1). Variances were homogeneous for each variable (Table 1). The lack of differences between years meant that data from 2015/16 and 2016/17 were pooled.

### 3.2. Associations with Total Hibernation Period

Total hibernation period was positively correlated with the number of arousals (*r_s_* = 0.40, *n* = 34, *p* < 0.01) and negatively correlated with the percentage of nights awake (*r_s_* = −0.35, *n* = 34, *p* < 0.05, Figure 2). This meant that the longer the hibernation period, the greater the number of arousals and the smaller the proportion of time spent awake, which prompted a further analysis for which we had no a priori hypothesis. Because the number of arousals increased and the proportion of time awake decreased, the longer the total hibernation period, we predicted that arousals would on average be shorter, the longer the total hibernation period. There was a tendency towards this prediction, but the relationship was not significant (*r_s_* = −0.22, *n* = 33, *p* > 0.1). None of the variables were normally distributed (Shapiro-Wilks, *p* < 0.05), so Spearman rank correlation coefficients were used.

### 3.3. Associations with Percent Weight-Loss

As predicted, the longer the total period of hibernation, the greater the percentage weight-loss (*r_s_* = 0.67, *n* = 21, *p* < 0.001, Figure 3a). Weight-gain during hibernation, demonstrated by four individuals, shows that hedgehogs can eat during arousals. This pattern remained when hedgehogs that gained weight were removed from the analysis (*r_s_* = 0.69, *n* = 17, *p* < 0.001), i.e., animals that gained weight by eating during arousals were not causing this pattern. Unexpectedly, percent weight-loss was not associated with the number of arousal events (*r_s_* = 0.12, *n* = 21, *p* > 0.60, Figure 3b), but there was a significant positive correlation between percent weight-loss and percent of nights asleep (*r_s_* = 0.66, *n* = 21, *p* < 0.01, Figure 3c). So, the proportion of time spent asleep in the hibernation period is more strongly associated with weight-loss during hibernation than is the number of arousal events. Variables were not normally distributed (Shapiro-Wilk, *p* < 0.05) or were percentages, so Spearman rank correlation coefficients were used in all cases. 

### 3.4. Associations with Pre-Hibernation Weight

There was a significant negative correlation between pre-hibernation weight of hedgehogs and the daily percentage weight-loss and with actual weight-loss; larger hedgehogs lost smaller proportions of their bodyweight each day than did smaller hedgehogs. This pattern was unaltered if hedgehogs that gained weight were excluded (Figure 4a, Table 2). There were no associations between pre-hibernation weight and total hibernation period, mean duration of arousal events or mean duration of intervals between arousal events (Table 2), but there was a significant positive correlation between pre-hibernation weight and the number of arousal events (Figure 4b, Table 2). So, the larger the hedgehog, the more often it wakes up during hibernation, but the less weight it loses. Variables were not normally distributed (Shapiro-Wilk, *p* < 0.05) or were percentages, so Spearman rank correlation coefficients were used in all cases. 

## 4. Discussion

Patterns of weight-loss and activity by hedgehogs during hibernation described in this study can inform over-winter husbandry of hedgehogs at rescue centres, for which there are no set or widely-used guidelines or recommendations about frequency of weighing and handling, nor robust evidence for how to anticipate weight-loss. We recommend that monitoring (e.g., frequency of weighing) should be tailored to individuals, particularly those likely to lose weight faster based on their pre-hibernation weight and hibernatory behaviour. 

### 4.1. Disturbance and Weighing

When comparing data from two winters, one with and one without weighing in mid-hibernation, there were no significant differences in the mean duration of total hibernation period, number of spontaneous arousal events, duration of these events or of the interval between these events (Table 1). This suggests that a single mid-hibernation weighing at the rescue centre did not influence important components of the hibernation. Of 21 hedgehogs weighed mid-hibernation in 2015–2016, only one woke soon afterwards and this may have coincided with a ‘normal’ spontaneous arousal. This suggests that concerns about effects of disturbance may be less serious than previously believed [8,27]. 

### 4.2. Spontaneous Arousals

The mean number of arousals per hedgehog fell within previously described ranges [19,24]. The mean and lower end of the range of number of nights between arousals was similar to those in Walhovd [19], but the upper end of the range (36 nights) was greater than elsewhere (15 nights [19]; 13 nights, [24]). The determinants of patterns of arousal are not yet known, but might include: differing methodologies used to observe these arousals (e.g., thermal measurements, visual observations, radiotracking); variation in environmental conditions; or variation in pre-hibernation mass or body-condition among studies. In the present non-invasive study, spontaneous arousal was only measured or recorded when an individual left their nest. This means that arousals that did not involve leaving the nest [8,19], could not be identified, and were not recorded. This could explain the apparently greater maximal number of nights between spontaneous arousals.

Although the mean duration of arousal events was typically <2 nights, some individuals woke for as long as nine consecutive nights, considerably longer than the 34–44 h reported by Walhovd [19] or the mean of 33.7 h (*n* = 6) observed by Kristoffersson and Soivio [24]. The latter did, however, observe one spontaneous arousal event lasting 121 h. Formal comparisons of published mean durations of spontaneous arousal with the present study cannot be made due to the different ways of measuring this variable. The long duration of some arousal events (up to 9 nights) could indicate that some individuals are not entering deep hibernation, perhaps due to conditions in the centre (food, nesting materials, disturbance) being different to those in the wild or to the relatively benign winter conditions in Cornwall. 

### 4.3. Weight-Loss

Spontaneous arousal is energetically expensive [18,30] and so more frequent switching between metabolic states of hibernation and arousal should add to the total energy ‘bill’ for the hibernation. The expectation was that the number of spontaneous arousal events would increase linearly with the length of the total hibernation period (as demonstrated here, Figure 2a). If their purpose is to prevent oxidative stress, reduced immunocompetence and neuronal tissue damage [17], longer hibernation would increase these risks and so more arousals are ‘required’ to offset physiological costs. The percentage of time spent awake decreased (Figure 2b) as hibernations got longer. A logical inference from this is that mean duration of arousal event should also be shorter when hibernations are longer, but the physiological basis for this is not clear. Whilst our data tend towards this pattern, there was no significant negative correlation. Larger sample-sizes from data collected over more years would provide greater statistical power when testing this prediction. 

As predicted, weight loss (a proxy for energetic cost) increased the longer the hibernation (Figure 3a). This strongly suggests that when making further comparisons between weight-loss and other variables (e.g., sex, body condition, starting weight, etc.) weight-loss should be presented as loss per day (g·d^−1^) rather than total weight-loss. In contrast with expectations, daily weight-loss was, however, associated more strongly with amount of sleep (Figure 3c) rather than the number of arousal events (Figure 3b). This may indicate that leaving and entering hibernation is not as energetically expensive as previously thought [18,30], but is perhaps more easily explained by the ready availability of food. In rescue centres, a small amount of food is provided throughout hibernation, so hedgehogs could feed each time they wake. The more often a hedgehog wakes and eats, the costs of arousal could be offset and less weight would be lost. Four individuals in this study even gained weight during their hibernation period (Figure 3). Such increases in weight may be explained by feeding during periods of arousal and/or taking on food in the time between final arousal and post-hibernation weighing. For reasons of certainty about final arousal, weighing post-hibernation can happen up to 6 days after waking, but is typically less than half this. Those that gained weight were also those that spent amongst the smallest proportion of their time asleep (Figure 3c), meaning that they had relatively more time available to feed during their hibernation. Although recording food consumption was not part of the original protocols, it merits further investigation as it may influence aspects of husbandry such as frequency of weighing required, and the amount of food provided during hibernation.

Body-weight is an important variable that helps determine types and amounts of medication prescribed by a veterinarian and administered by rescue centres to unwell hedgehogs. It also influences whether healthy hedgehogs should be allowed to enter hibernation and whether they should be released [5]. We expected there to be some association between pre-hibernation weight and five different characteristics of hibernation (Table 2). We present evidence that hedgehogs of different weight do hibernate differently. For instance, the number of arousal events increased with pre-hibernation weight (Figure 4a), but length of hibernation did not (Table 2). If larger hedgehogs need to wake more often during hibernation, then they may be developing relatively more physiological costs than smaller individuals and therefore need to off-set these costs with more regular arousals. This is in contrast with broad recognition that metabolic rates scale inversely with body-size [31]; however, it may be that hibernation creates different physiological costs than just calorific expenditure for arousal. 

If weight is an underlying determinant of arousal, a regression (using a larger sample over multiple years) of number of arousals against pre-hibernation weight would allow prediction of how often hedgehogs will wake. Although not strictly statistically rigorous (due to values of the x-variable not being fixed), this could be used to anticipate the total (and different) amounts of food required for each animal (for potential consumption during arousal). 

Daily weight-losses (mean ± s.e.; 1.18 ± 0.48 g·d^−1^) fell within Wroot’s [14] values of 0.8–2.0 g·d^−1^, but our range (3.5 to −6.2 g·d^−1^) was much greater. Daily weight-loss in the present study (in terms of percentage and absolute amounts) were negatively related to pre-hibernation weight (Figure 4b,c). So, larger hedgehogs woke up more often (Figure 4a), but lost less weight than did smaller hedgehogs (Figure 4b,c). This is the opposite to previous studies [9,11,32], which showed that individuals that had more mass to lose tended to lose more mass. This unexpected pattern is, again, perhaps best explained by the ready availability of food. Heavier individuals wake up more often and can, therefore, feed more often, meaning that they lose less weight, highlighting the need to monitor food consumption during captive hibernation. 

### 4.4. Captive vs. Wild Hibernation

Conditions for hibernation in captivity are clearly rather different to those in the wild. Captive hedgehogs have no ability to move nests or select different nesting materials (as are observed in wild hedgehogs [9,15,21]. In the wild, animals have the opportunity to move around and forage during periods of arousal, but food may not be readily available because it is wintertime. Although captive hedgehogs can move around their hutch when awake, they cannot move the distances observed for wild hedgehogs. Decreases in food availability may well be one of the cues for hibernation [8], and moving nests may consume considerable energy. Thus, artificial conditions where food is plentiful and movement is restricted may cause hibernation by juveniles in captivity to be different from those in the wild. In particular, the duration and depth of hibernation may be less. Fruitful areas of research might include regional variation in duration and initiation of hibernation and how these may respond to climatic change. There is already evidence that the ‘trigger’ temperature for hibernation varies across Europe [8] and unseasonally mild conditions, potentially the consequence of a warming world [26], can delay the onset the hibernation [9,15]. For example, when the Autumn of 2014 was far milder than normal, juvenile hedgehogs in Denmark initiated hibernation a month later than shown in previous studies [9]. In the present study, durations of hibernation were similar between years (Figure 1a), but the maximal hibernation period was much shorter in the 2016/17 winter (70 nights) than in 2015/16 (111 nights). This may have been a consequence of the warmer mean minimal temperatures in February and March 2017 (6.1 °C and 8.0 °C) than in the previous year (5.1 °C and 5.1 °C; [33]). Unseasonably warm periods during hibernation may stimulate arousals and additional energy expenditure [34]. With increasingly mild winters [35], and unseasonably mild episodes in winter, the occurrence of multiple distinct periods of hibernation per winter, separated by abnormally long arousal events appear to be more evident. It would be useful to evaluate whether these behaviours occur more broadly. Long-term changes in the timing and duration of hibernation may require physiological and behavioural adaptation as well as phenological adjustments to altered availability of prey items. Such changes could have implications for rehabilitation methods and practice in order to maximise both welfare and its contribution to species conservation.

### 4.5. Recommendations for Overwinter Monitoring

The implications of these patterns observed here for over-winter husbandry of hedgehogs in rescue centres are several. We recommend closer monitoring of arousals and of food consumption during arousals. 

Provision of food: Availability of food in captivity appears to reduce the amount of weight lost during hibernation; at least, those individuals that woke (and thus could feed) more often lost less weight per day than those that woke less frequently. If arousal is energetically expensive [18,30], but necessary for other physiological reasons [17], then these costs can be offset by eating. Food should certainly be made available for hedgehogs hibernating in captivity for consumption during periods of activity, to help offset weight-loss and improve likelihoods of surviving hibernation. The possibility that availability of food is encouraging shorter hibernations and more frequent, longer arousals requires greater investigation.

Monitoring weight and behaviour: The lack of evidence that mid-hibernation weighing causes disturbance could be interpreted as meaning that this level of monitoring should not be of major concern for the welfare of overwinter hedgehogs and supports the notion of weighing at intervals more frequent than 6 weeks. This could be particularly valuable for smaller hedgehogs (that lose weight faster than larger individuals) that may be approaching weights (e.g., ≤400 g) where the risk of not surviving hibernation is greater. This approach would benefit from further studies to test for any cumulative effects of multiple disturbances. Durations of hibernation vary amongst individuals, so weight-loss is better presented as g·d^−1^ or %.d^−1^ rather than total loss.

Weights and duration of hibernation should not be the only variables to be monitored during over-winter care of hedgehogs at rescue centres. Frequency and duration of spontaneous arousals, indicated by disturbance outside the nest area, are easily and non-invasively observable, without disturbing the animal. Unfortunately, recording consumption of food was not part of the protocol that provided the data for the present study. Given the negative association between weight-loss and number of arousals, knowing how often and how much food is consumed would have been informative. Individuals waking regularly but not eating may well lose weight much faster than those that do eat and maybe even faster than those that wake infrequently. Such individuals may benefit from more frequent weighing to ensure they are artificially woken if they approach the minimal threshold for weight. Protocols for observations should be expanded to include the routine collection of data on frequency and duration of arousals and consumption of food, thereby allowing comparisons among locations, sex and prior conditions (weight, disease, medication, etc.). Further investigation is required to determine how these multiple variables influence hibernation.

## 5. Conclusions

Rescue centres can collect large amounts of data and have the potential to create evidence-based monitoring protocols intended to improve welfare of patients and success of rehabilitation. Key points that were demonstrated include: (i) mid-hibernation weighing did not seem to affect hibernation; (ii) weight-loss increased with duration of hibernation, but appeared more strongly associated with the proportion of time spent asleep than with the number of arousal events; (iii) in contrast with previous studies, larger hedgehogs lost less weight per day than did smaller hedgehogs; (iv) mean values for components of hibernation were similar to values recorded from individuals in the wild, but extremes for weight-loss (or gain) and duration of arousal events were much greater for animals being managed in a rescue centre. Much of (ii) and (iii) can be ascribed to plentiful availability of food. Differences in hibernatory behaviour between captive or wild hedgehogs may well be due to a range of non-natural conditions in hutches. There is increasing evidence that milder winters associated with climatic change are changing the ways in which hedgehogs hibernate. This has uncertain implications for the long-term future of declining populations. The patterns described here provide much-needed information for rescue centres caring for hedgehogs and highlight areas for further research. Hibernation protocols for rescue centres should be updated following the recommendations we make. We hope that this will allow more successful rehabilitation which will, in turn, help support populations of this charismatic, threatened species.

## Figures and Tables

**Figure 1 animals-10-01418-f001:**
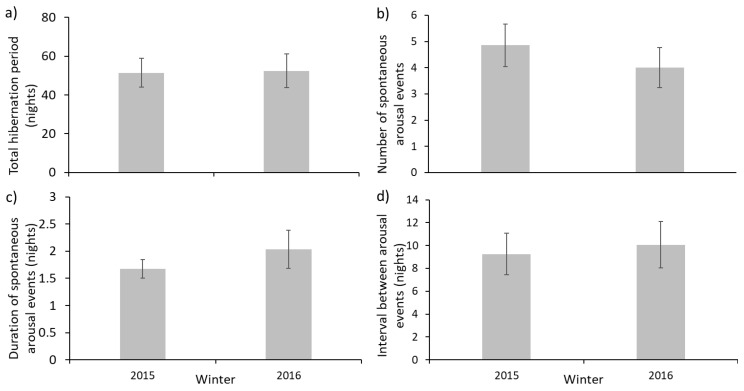
Mean (±s.e.) components of hibernation by hedgehogs in Cornwall between the winters of 2015–2016 (*n* = 21) and 2016–2017 (*n* = 13); (**a**) total hibernation period, (**b**) number of spontaneous arousal events, (**c**) duration of spontaneous arousal events, (**d**) interval between spontaneous arousal events i.e., number of consecutive days asleep.

**Figure 2 animals-10-01418-f002:**
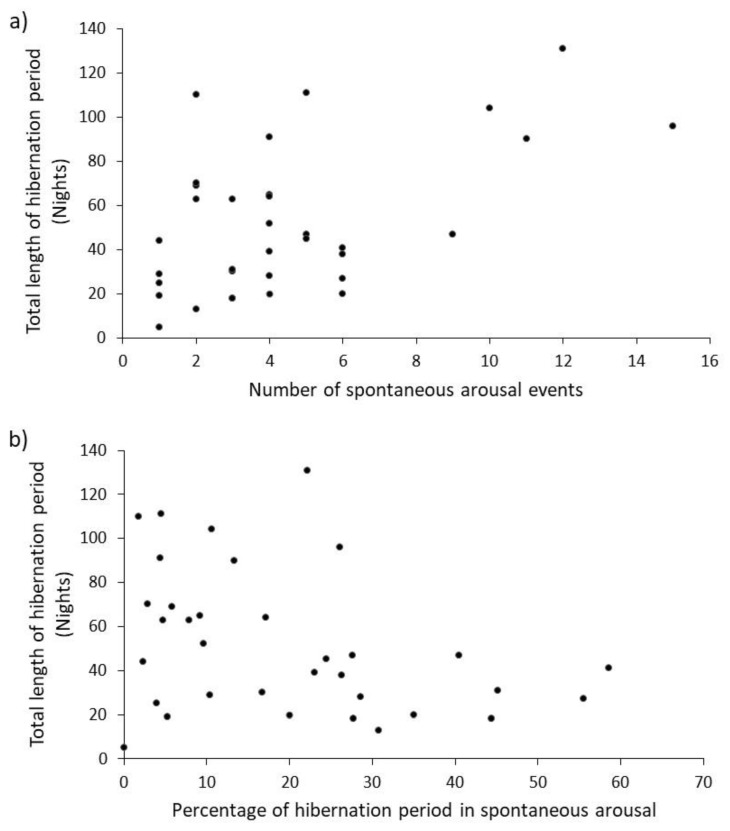
Correlations between total hibernation period and (**a**) number of spontaneous arousal events and (**b**) percentage of nights spontaneously aroused (*n* = 34).

**Figure 3 animals-10-01418-f003:**
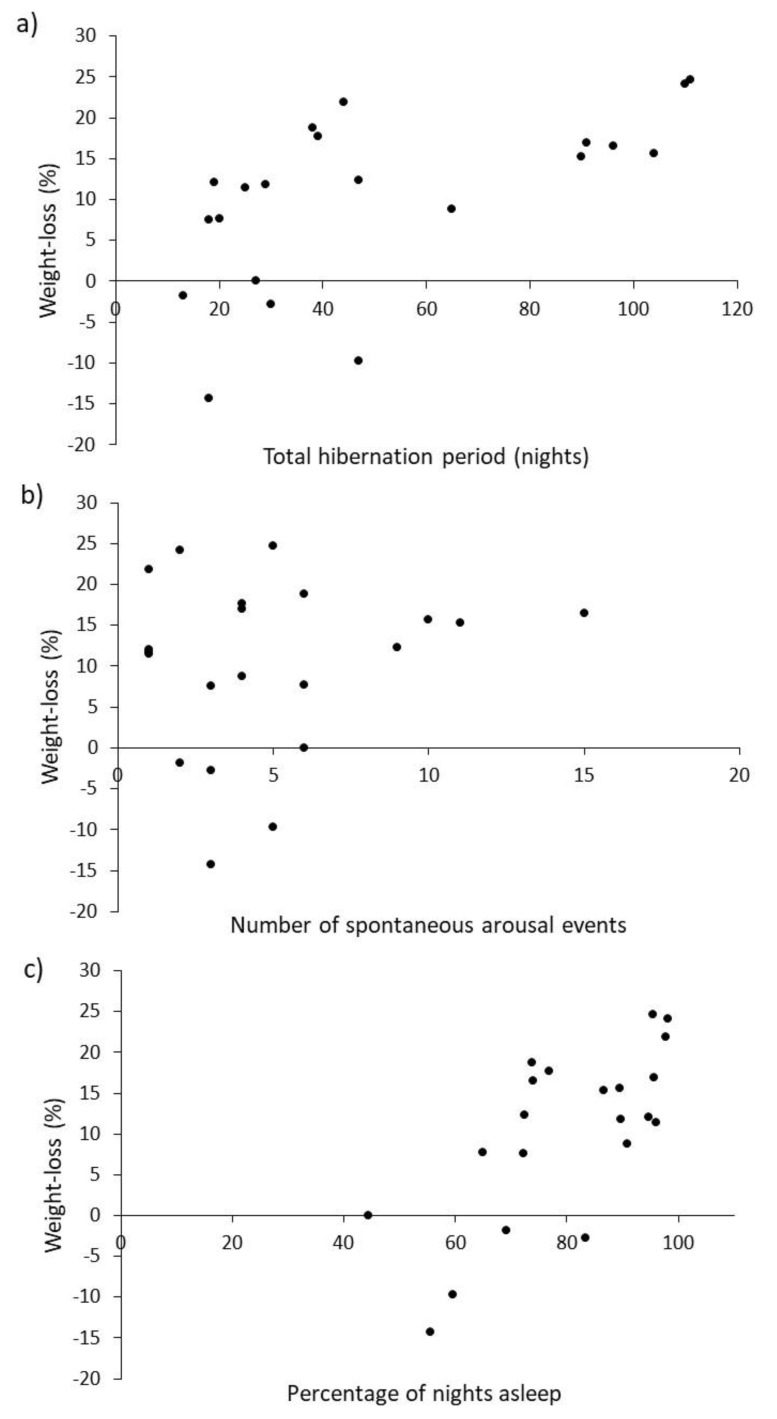
Correlations between percent weight-loss and (**a**) total hibernation period, (**b**) number of arousal events and (**c**) percentage of nights spent asleep (*n* = 34).

**Figure 4 animals-10-01418-f004:**
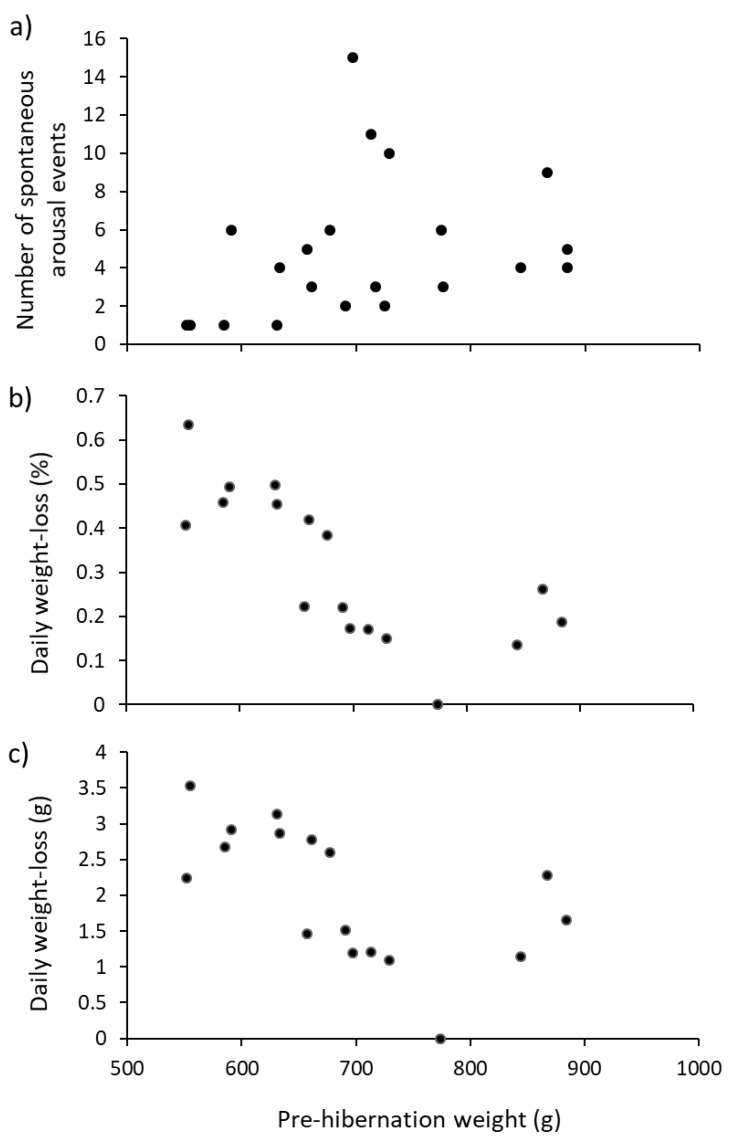
Correlations between pre-hibernation weight (g) of hedgehogs in the winter of 2015/16 (*n* = 17) and (**a**) number of arousal events, (**b**) percent daily weight-loss and (**c**) actual daily weight-loss (with (**b**,**c**) excluding individuals that gained weight).

**Table 1 animals-10-01418-t001:** Fisher’s F tests for heteroscedasticity and Student’s *t*-tests to compare means for total hibernation period, number of arousal events, duration of arousal events (nights) and number of nights between arousal events between the winters of 2015/16 and 2016/17.

Variable	Fisher’s *F* Test			Student’s *t*-Test		
	*F*	d.f.	*p*	*t*	d.f.	*p*
Total hibernation period	1.20	20, 12	>0.70	−0.08	32	>0.90
No. of arousal events	1.80	20, 12	>0.20	0.72	32	>0.40
Duration of arousal events	0.39	20, 12	>0.05	−1.04	32	>0.30
Duration of interval between arousal events	1.31	20, 12	>0.60	−0.28	32	>0.70

**Table 2 animals-10-01418-t002:** Correlations between pre-hibernation weights and daily weight-loss, total hibernation period, number and durations of arousal events and the duration of intervals between arousal events. Significant results are in italics.

Correlation of Pre-Hibernation Weight with	*r_s_*	*n*	*p*
*% daily weight-loss (incl hedgehogs that gained weight)*	*−0.80*	*21*	*<0.001*
*% daily weight-loss (excl hedgehogs that gained weight)*	*−0.80*	*17*	*<0.001*
*Actual daily weight-loss (excl hedgehogs that gained weight)*	*−0.67*	*17*	*<0.01*
Total hibernation period	0.24	21	>0.2
*No. of arousal events*	*0.45*	*21*	*<0.05*
No. of nights per arousal event	0.23	21	>0.3
No. of nights between arousal events	−0.30	21	>0.1

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
