# Peer review of "Hibernation Patterns of the European Hedgehog, Erinaceus europaeus, at a Cornish Rescue Centre"

_animals, 2020, doi:10.3390/ani10081418_

Round 1

Reviewer 1 Report

I apologise for all the comments which I made as I went through the paper. They will hopefully provide some ideas for rewriting.

Main text

Line 17: Change in font size

Line 21: Unsure how the findings can really influence over winter care

Line 41: The introduction is a bit rambling and not always relevant to the research

Line 47: Can ‘parasite burdening’ in itself be a reason for admission?

Lines 51-54: There more recent references on the physiology of hibernation.

A definition of hibernation and what is physiologically happening is required                     – some of this information is included but is spread over the introduction

Line 55: Is hibernation itself a mortality factor? What are they dying of?

Is reference 8 a chapter in this book? If so need chapter title, author etc.

Line 57: I’m not sure reference 5 is based on published data, there are better                                 references in the literature.

Lines 68-81: beyond the key facts this is probably more ‘discussion’ than ‘introduction’

Line 87: Did you consider types of nesting material?

Lines 89-91: Not sure how the research has improved husbandry overwinter. Other research would also suggest that these animals should just be being                          released.

Line 122-135:  The hypotheses are hard to follow as they are about the statistical tests carried out. It might be better to word this section in real and practical terms.

Line 139: Unclear what the protocols of the rescue centre were

Line 142: Unclear how environmental conditions could be ‘consistent for all                                     individuals’. If they were outside would temperature potentially not vary                               considerably and potentially day length over the study period?

Bedding  was shredded  paper (an unnatural material) but the introduction                       suggests bedding type is a factor in hibernation -this needs to be discussed.

Line 143: The hutch sizes seem too small to mimic the distances that wild hedgehogs would move between nests in the winter – needs discussion

Line 145: Not enough information about the individual hedgehogs e.g. why admitted to the centre, any medication given, if clinically healthy at the start of the study, age, sex, size (e.g.girth or length), body condition score and/or weight at admission

Line 149: Unclear as to the significance of green faeces (these may arise for a variety of clinical reasons).

Line 152: Did this result in arousal?

Line 153: Why 400g?

Line 154: If hutches were checked daily, might this alone not have resulted in                                   arousal?

Line 157: Why was food eaten not recorded? This seems really important if considering weight changes.

Line 158:  It is not really clear what the management differences in the second year were. The methodology is generally a bit long winded and unclear.

Line 162: The criteria for suitability for hibernation and inclusion in the study needs to be earlier

Line 224: Hedgehogs DO eat during arousal. Monitoring food intake would have been useful.

Lines 254-258: Unsure how the findings really improve protocol and provide guidance for overwinter care.

Lines 267-268: I don’t think those are minimum winter temperatures are they? They seem very high even for Cornwall!

There could be discussion around the effects of temperature  on hibernation here, including references to published work

Lines 269-277: Not enough is made of the fact that these are not natural conditions. The hedgehogs cannot move very far, don’t have the opportunity to change nest or forage naturally and food is provided. The contrast with the wild should be discussed in order to validate the data and explain why it is useful despite being in captivity.

Lines 284-299: This is all OK, but ignores perhaps the other influencing factors like temperature, age and weight of  hedgehog, sex etc. This is all multifactorial.

Line 298: You’d want to PM hedgehogs that were ‘healthy’ to have a comparison rather than those dying of other things

Lines 300-308: Not measuring food intake was perhaps a mistake. This section needs to discuss what is known about feeding during arousal periods in the wild.

Lines 309-301: The key here is are captive hedgehogs behaving in the same way as wild ones?

Line 322: Does ‘treatment’ here mean the decision to allow an animal to hibernate or not?

The emphasis is all on body weight. How else did these hedgehogs differ pre                    hibernation (e.g. age, sex, prior disease) might these have also been                            influencing factors?

Line 342: The amount of food would be useful to know, but the study didn’t measure this or the amount consumed.

Line 349: The concept of hibernation being ‘safe’ is an odd one, especially if food is provided. Hedgehogs in the wild don’t have someone to wake them up if they drop below 400g.

Line 375: It is still perhaps unclear what the wildlife rehabilitator has learnt from this study and how it might change the way they practically manage hedgehogs.

References

  • The literature review for this article need to be improved. There’s lots more relevant published work around this subject. Would suggest that Yarnell et al., 2018 (article reference 6) is a useful example of how some of the points noted above might be covered and also includes some useful up to date references. Some references included are also quite old (over 10 years old) this should not be the case unless they are really key references.
  • Some inconsistence in reference lay out e.g. not all dates in bold.
  • Reference 8 is a book chapter I think, the reference needs the chapter title and author details. The editors are Mullineaux and Keeble.

Reviewer 2 Report

South, Haynes and Jackson provides an interesting study on the hibernation of European hedgehogs, which confirms previous results and assumptions. This is very useful. Especially the new information on how arousals seem to cause less weight loss than time spent sleeping is interesting, as it has previously been assumed that the arousals were much more energy expensive.

However, we should keep in mind that this study has been done on hedgehogs in captivity. So the more sudden arousals when e.g. a predator attacks the hibernaculum or the hibernaculum is disturbed by humans doing gardening, is not recorded here, and the question is whether they would be more energy demanding. The individuals of the study may have already become habituated to the handling by humans, so would this count as a “true” disturbance? And as previous studies have recorded up to eight nest changes per individual during the hibernation period, it is also important to note this as a limitation to the present study, as it does not mimic the natural conditions, because the hedgehogs hibernate in a rehabilitation centre in small cages. These individuals are in captivity and may therefore also experience higher levels of stress and perhaps not show a completely natural behaviour.  I would really like to see this all mentioned in the discussion. And I also recommend that you perhaps include the duration of admission to the centre as a factor in your analysis as stress and/or habituation may very likely influence your results.

L. 17-18+ 34-35: There is something wrong with the format of this text, at least in the version of the manuscript I have received.

L. 49: Sometimes? I am not sure I agree here.

L. 49 and 57 and 60: I am not aware if it is the type setting gone wrong, but there are large gaps between . and the first word in the next sentence. It happens throughout the manuscript.

L. 56: I wonder why you have not cited the most recent study on juvenile hedgehogs and hibernation including weight loss, nest changes, survival (Rasmussen, SL,  Berg, TB,  Dabelsteen, T,  Jones, OR.  The ecology of suburban juvenile European hedgehogs (Erinaceus europaeus ) in Denmark. Ecol Evol.  2019; 9: 13174– 13187. https://doi.org/10.1002/ece3.5764)

L. 73: There is a huge gap between “adding” and “50 g”.

L. 82: The previously mentioned article does actually contain more data on this.

L. 149: How did you make sure the green faeces was not indicating disease instead? And how did you in fact make sure that all hedgehogs entering the study were not sick? Could you perhaps make a small description of this?

L. 153: What treatment would an individual below 400 g and with the arousal induced, get? And for the results: did any of the individuals die? Did any of the individuals go below 400 g and becoming exposed to an induced arousal? And what happened then?

L. 158: What kind of disturbance are we talking about here, other than the weighing? And did this only take place once after the 6 weeks of hibernation?

L. 165: Perhaps it would be beneficial to describe how it was evaluated that the individual did not leave the nest? (I guess it is the same as in line 155).

L. 201: When was it appropriate to pool data and when was it not?

L. 204: In my opinion the titles on the x-axes are a bit confusing, as they are almost the same (length of arousal events, total hibernation and interval between arousals) and the explanation can only be found in the figure text. Could they perhaps be specified a bit on the figure?

L. 208: Perhaps it is just me, but I find the Table 1 a bit confusing, since it is not intuitively clear to me at the first glance which result belong to the Fisher’s F tests for heteroscedasticity and students t-test.

L. 234: Should it be explained somewhere how you define % weight loss? I guess the negative values indicate individuals gaining weight during hibernation?

L. 239: Now I am getting confused. You have a range of definitions of weight loss, but as I read it here (opposed to the abstract) larger hedgehogs lost less weight than smaller ones, also in general (actual weightloss)? And if I am not right, where do you describe the result showing that larger animals loose more weight during hibernation?

L. 246: You have forgotten the e in hedgehogs in table 2

L. 188-252: In general you tend to describe positive or negative correlations of your data. Sometimes you describe what it actually means, and other times you don’t. Keeping in mind that many rehabilitators will probably want to read and understand this article, I think you should consider explaining what it actually means. E.g. “there was a significant positive correlation between pre-hibernation weight and the number of arousal events (Figure 4 b, Table 2).“ What does this mean in reality? (I know that we do not write scientific articles for laymen, but this article will likely be pretty important information to the rehabbers, which is why we should do our best to communicate the results in a way most people would understand).

L. 318: You give some possible explanations for why an individual would be awake for e.g. 9 days during hibernation, but honestly, what is that hedgehog doing in a cage in a rehab centre if it stays awake for so long (and is well)? Wouldn’t it be suitable to perhaps add to the discussion that one could consider releasing such an individual if the conditions are adequate and the individual shows no sign of disease?

Do you have any potential explanation for the fact that some only hibernated 13 nights and 5 nights (l. 192)? And what happened to these individuals? Were they sick? Or were they just awake and active in the cage? And did you release them soon after?

L. 322-335: I think you should really consider including a discussion on weight vs body condition. There seem to be so much focus on weight, but a juvenile of 600 grams should be in very good condition and a large adult of 600 grams could be in extremely poor body condition.

L. 347: Am I completely lost here? Smaller individuals losing more weight? In theory? In other species? Or in your study, and is it percentagewise or actual weight loss? I thought the abstract said the larger individuals lost more weight?

L. 357: adaptation to climate change… as seen in Rasmussen, SL,  Berg, TB,  Dabelsteen, T,  Jones, OR.  The ecology of suburban juvenile European hedgehogs (Erinaceus europaeus ) in Denmark. Ecol Evol.  2019; 9: 13174– 13187. https://doi.org/10.1002/ece3.5764

L. 370: Again… percent weight-loss by hedgehogs during hibernation was greater the lighter the pre-hibernation weight.

L. 382: It seems this part (Funding) has not been written?

L. 389: It seems this part (Conflicts of interest) has not been written

Round 2

Reviewer 1 Report

Amazing! This is so much better for the changes made. Really well done and thank you for your patience and feedback with the editing process.

A few minor comments:

Line 4              Does this need little superscript numbers on the authors’ names to reference to their contact details?

Line 61            Space missing before the last references

Line 77            Should weight.day-1 be weight day-1 as in line 68? Not sure.

Line 90            Not sure what this journal wants but I’d usually use ‘and others’ rather than ‘et al’ when written in the main text. I may be wrong here, but perhaps check.

Line 107          As above

Line 133          Sounds a bit overstated, maybe go for: ‘Conditions in captivity are however, clearly different to those in the wild’

Line 147          I’d not refer to the supplementary material here, I’d include it in methods (as you’ve done in line 176)

Line 172          Do the RSPCA actually ‘supervise’ you? or do you just ‘work closely’ with them? Do they still do centre approval? are you one of those?

                        Should your veterinary practice not get a mention too, especially as you are including a table with prescribed drugs in it? Even if you just said you ‘worked closely with a local veterinary practice’ or similar.

Line 176          Think you have to say your medication was veterinary prescribed as it includes POM-V drugs. Go for something like ‘…appropriate medical treatments, as prescribed by the centre’s veterinary surgeon, were administered………’

Supplementary material 1 – attach as a PDF rather than an Excel sheet.

Line 179          Refer specifically to the Protocol. So maybe something like: ‘No individual was considered for use in the study and allowed to hibernate unless it fulfilled the conditions of the centre’s Hibernation Protocol (Supplementary material 2) including having all medications completed, being >600g in weight and in good body condition.’

Supplementary material 2 – do something to it so it fits on one side of A4 and attach as a PDF rather than a word document.

Line 182          The at al question again

Lines 188-190 I don’t think BSAVA really ‘approve’ hedgehog bedding. Maybe change to something like: ’Paper, whilst different to natural nesting materials, is readily available, dust-free, widely used in British rescue centres and recommended by other authors

[7].’

Line 195          The green faeces, maybe: ‘…..healthy hedgehogs, they are recognised as an indicator of…’ or ‘…. they may be an indicator of…”. Sorry I know I’m being difficult, and you know what you mean, but I just don’t want people to read this and think green faeces= hibernation.

Line 203          You can remove the word ‘suitable’

Line 204          Do you mean ‘carnivore’ here rather than ‘insectivore’? I assume you mean cat food? Readers in other parts of the world where insectivorous diet is maybe cheaper might wonder. You could include the make of biscuits in brackets here to be clear.

Line 310          Remove ‘perhaps’, be confident in your suggestion (you could go for ‘recommend’ on line 309 instead of  ‘suggest’)

Line 369          6 days

Lines 375-376 The start of this sounds like rescue centres are perhaps selecting medication, which would potentially be illegal. Change to something like: ‘Body-weight is an important variable that helps determine types and amounts of medication prescribed by veterinary surgeons and administered to sick hedgehogs by rescue centres………’

Line 398          At the end of the line, maybe reiterate again that any further studies should monitor food intake.

Line 420          Where you say “With increasingly mild winters [34], and unseasonably mild episodes in winter, the occurrence of multiple distinct periods of hibernation per winter separated by abnormally long arousal events are becoming more apparent (this study; K. South, pers. obs.).” is this both in this study and a personal observation? Do we need both (or either) as you are the first author and this is discussion? Could you include ‘appear to be’ in there somewhere? e.g. ‘With increasingly mild winters [34], and unseasonably mild episodes in winter, the occurrence of multiple distinct periods of hibernation per winter separated by abnormally apparent appear to be more apparent.’

Line 431          Be more assertive and change to: ‘We recommend closer monitoring …..’

Line 433          I’d pull line 434 up here after the colon – but may depend on journal style

Line 443          I’d pull line 444 up here after the colon - ditto

Line 448          (e.g. ≤ 400g) or (e.g. < 400g)?

Line 452          Weights and duration of hibernation? and arousal?

Reviewer 2 Report

Great work on the revision of the manuscript. 

Author Response

Thank you for your valuable advice and guidance.